# The Role of Policy Perceptions and Entrepreneurs' Preferences in Firms' Response to Industry 4.0: The Case of Chinese Firms

**Chenguang Li** [1,*] **, Zhenjun Qiu** [1] **and Tao Fu** [2]

1 School of Economics and Management, North China University of Technology, Beijing 100144, China; chilqzj@mail.ncut.edu.cn
2 School of Economics and Management, Beijing University of Technology, Beijing 100122, China; futao@bjut.edu.cn
* Correspondence: lichenguang@ncut.edu.cn; Tel.: +86-010-8880-2792

**Abstract:** Favorable policy implementation results are due not only to policy makers' abilities but also to the behavior of those responding to the policies. For example, a CEO's understanding of a government policy's content and his or her willingness to respond to it, based on the expectation of profits, plays a vital role. To understand the relationship between how policies are perceived and how enterprises behave in response to innovation policies in the era of the 4th Industrial Revolution (Industry 4.0), in this study, we use structural equation modeling to investigate the roles of various factors and examine the response mechanisms in enterprises through which entrepreneurs react to Industry 4.0 innovation policies. The hypothesized model is validated empirically using a sample collected from 337 domestic Chinese high-tech firms. The modeling results indicate that positive perceptions of policies have a positive effect on entrepreneurs' preferences which, in turn, motivate positive behavior toward innovation. Moreover, testing the model showed partial and complete mediation effects, indicating that the perceived practicability of a policy is a factor with a strong impact on response behavior that sometimes exerts its influence by altering the mediator of entrepreneurs' responsive preferences. The empirical results and management implications of this study can serve as a reference for the effective implementation of and response to government development plans.

**Keywords:** Industry 4.0; innovation policy; policy response; policy perception; entrepreneurs' preferences

## 1. Introduction

The 4th Industrial Revolution (Industry 4.0), which has attracted much interest in recent years, has brought new development opportunities to the manufacturing and service industries [1–3]. In Germany, the phrase "Industry 4.0" is used to describe the digital transformation in manufacturing. However, the concept is understood differently in different countries, where "Industry 4.0" can refer to a key tool for implementing the national strategy of innovative development (USA), the leading sphere of industry (UK), the modern industrial reform (France), the plan for scientific and technological modernization (Japan), and manufacturing innovation and transformation (China) [4].

"Made in China 2025", with the sustainability concepts of "innovation, coordination, green, open, and sharing", is China's Industry 4.0 national strategy. It aims to endorse the importance of research and innovation and is of revolutionary significance in industry. As one of the major countries in the world and the leading developing country, China's manufacturing sector has been challenged for some time by rising labor costs, environmental and resource difficulties, and a slowdown in exports. Made in China 2025 will attack these problems by using mandates, subsidies, and other methods to persuade manufacturers to upgrade their factories to become more competitive, innovative, and efficient—in short, enabling China to become a pioneering, high-end manufacturing power [5]. In China, political connections between firms and the government are quite common. However, such

political connections are not limited to government-owned firms, and some private firms may also have CEOs and/or directors with strong political backgrounds [6]. Moreover, Chinese high-tech firms remain highly dependent on the national government for institutional support and critical resources such as bank loans [7,8]. Thus, high-tech enterprises without political connections have strong incentives to respond to government policies in order to gain access to factor and capital resources critical to firm growth. From an empirical viewpoint, China is a particularly interesting case to analyze because of its fast-growing economy, as well as its controversial imitative activities in the international market [9]. The diversity in China's institutional environments allows us to observe interesting variations in some of the specific factors potentially relevant to the relationship between policy perceptions and high-tech enterprises' response behavior toward innovation.

Classic innovation theory regards policy factors as external influences on innovation and also believes that entrepreneurs' decision making is restricted by external factors [10]. Independent management of enterprises and policy intervention has become the focus of academic debate. The theory of the "national innovation system" emphasizes the stimulating effect of the government, and Lundvall argues that innovation policies should provide support and protection for implementation and redistribution of resources, and suggests that the response to a policy is more likely to be positive if there is more active internal research and development in the system [11]. However, Porter's "Diamond Model" emphasizes that national and regional competitive advantage come from the willingness and ability to engage in technological innovation, highlighting the dominant role of enterprises in generating competitive advantage and pointing out that policies are simply the providers of the resources enterprises need and the builders of the innovation environment, whereas entrepreneurs need to choose the best strategy as they evolve dynamically in pursuit of development opportunities [12]. Since then, entrepreneurs have been considered "rational actors" in terms of how they set their decision preferences. Davis proposed the Technology Acceptance Model based on rational behavior theory, emphasizing that behavior is determined by intentions, and how attitudes are translated into behavior is determined by the perceived usefulness and perceived accessibility of a course of action [13]. Ashford proposed that policy response behaviors represent the feedback of enterprises' support (opposition), implementation (resistance), etc., to the policy and that enterprises are independently selective in terms of how they respond to innovation policies and gain access to resources [14]. However, our understanding of policy perceptions and entrepreneurs' preferences regarding how to obtain external resources for innovation is not comprehensive. In recent years, much attention has been paid to responsible innovation theory, which emphasizes technology assessment, system innovation management, and innovation risk reduction based on system innovation [15]. Stilgoe et al. [16] place more emphasis on the fact that the response is the enterprise's reaction to the dynamic environment based on the expectation of results in terms of innovation. Therefore, the rational behavior of an enterprise's response to Industry 4.0 policies described above reflects the fact that the response to a policy results from the enterprise's perceptions of the policy's accessibility and practicability, its preferences in terms of how responsive it is, and its behavior concerning transformation through innovation; this understanding of the process is suitable for in-depth research of a policy application acceptance model [17].

Most governments have deployed various Industry 4.0 policies and instruments to foster innovation in enterprises [18–20]. However, there is some controversy regarding the effectiveness of this type of innovation policy. Supporters of such policies find evidence that they have a positive impact on innovation in enterprises [21–23], whereas their opponents claim that such policies often fail [24–26]. As market failures in the innovation system emerge, policy interventions are introduced in response [27]. An enterprise's response to Industry 4.0 policies is an important driver in its quest to obtain external resources and stimulate its innovative vitality. In the process of responding, the perception of the utility of the policy and the entrepreneur's response preferences are often neglected, leading to

a fuzzy understanding of the factors behind these responses and of how the behavioral paths concerning innovation intentions are influenced by policy [28]. Exploring response paths and key influencing factors is a real problem that urgently needs to be solved.

Concerning how enterprises respond to innovation policies, some scholars, taking a rational behavior perspective, propose the influence of policy perceptions and innovation intention on behavioral intentions [17,29]. Other scholars start from the process of innovation, emphasizing the influence of demand cognition and management responsive preferences on response behaviors [30]. However, the response to an innovation policy is a complex adaptive system [31]; in the "stimulus–response" process of supply and demand matching, the design of policy content [32], response threshold [33], and actual utility [34] should be considered. Taking these factors into consideration is conducive to a comprehensive understanding of responses to Industry 4.0 policies.

Hence, this article investigates the following three research questions: (1) Among the degree of difficulty of responding to the policy, its utility, adaptability responsive preferences, and enterprises' response behaviors, which are the factors influencing the process of enterprises' responses to policies? (2) Are there any intermediary factors in the response process? (3) What is the key response path? All these questions need to be examined and answered from a new point of view.

The aim of this study is to understand the relationship between perceptions of policies and enterprise response behavior concerning innovation in the era of Industry 4.0. We introduce the Policy Acceptance Model [17] to combine the existing policy response theory with the complex adaptive system theory and explore the policy response mechanism from a stimulus–response perspective. First, based on theoretical analysis, we build the model of how enterprises' responses to Industry 4.0 policies are influenced by perceptions of these policies and entrepreneurs' preferences, and explores the response path theoretically. Then, taking high-tech enterprises in China as a sample to collect relevant data, we use structural equation modeling to verify the impact of perceptions of policies and entrepreneurs' preferences on policy response in order to provide a decision-making basis for the government, enterprises, and relevant institutions. The study not only expands the application of Technology Acceptance Model in the policy field theoretically but also provides a useful reference for a mechanism for optimizing creative policy formulation and entrepreneurs' response decisions.

The rest of this article is organized as follows. The following section develops the research hypotheses. Section 3 summarizes our research methodology. Section 4 presents the results and analysis. Section 5 discusses our findings and their managerial implications. The last section concludes the paper with limitations and future research directions.

## 2. Hypotheses Development

### 2.1. Policy Perceptions and Entrepreneurs' Responsive Preferences

Previous studies of innovation policies tend not to be concerned with how policies are perceived in terms of their accessibility and practicability. However, empirical studies in some countries show that enhancing policy design and ease of responsiveness strongly contributes to the innovation of enterprises and the growth of the national economy. Moreover, the level of ease of responding to an innovation policy in an enterprise determines the effect of implementing the innovation policy [32]. For example, the importance of the implementation subject, application scope, participating institutions, and fund management content is clear from the experience of the SBIR and STTR programs in the United States, where strong targeting and clearly supportive policies in the two stages helped enterprises respond easily and achieved good results [35]. China's financial, tax, and fiscal policies also have had a positive effect on improving and stimulating technological innovation, and enterprises respond more actively to these kinds of policies because they are highly pertinent and easy to understand and respond to [22]. The function of an innovation policy is to enable technological innovation and product innovation to meet market demand and to promote technological research and development to the point of having developed a final

product, as well as producing high value-added economic benefits and social influence [36]. However, in the process of its establishment and implementation, the innovation policy tends to fall flat because of the lack of systematic cognition, market information, dynamic changes of the intervention time, and other reasons. In addition, the innovative activities that the innovation policy addresses are usually risky, and the policy response process is full of uncertainty; thus, for participants to rely on a "foresight" enterprise culture, correctly interpreting policy content and reaching consensus with key stakeholders, has become a necessary condition to reduce the risks and uncertainties [34], meaning that entrepreneurs' innovation management ability is closely related to their success in innovation. Thus, there is greater pressure on entrepreneurs to pursue more transparency regarding the support information and the response threshold of external government policy. Therefore, we propose the following hypotheses:

**Hypothesis 1 (H1).** *The perceived accessibility of a policy has a positive effect on its perceived practicability.*

**Hypothesis 2 (H2).** *The perceived accessibility of a policy has a positive effect on entrepreneurs' responsive preferences, i.e., the easier the policy is to understand and respond to, the deeper the entrepreneurs' sense of identifying with the policy, and the more likely it is that they will accept the policy.*

**Hypothesis 3 (H3).** *The perceived practicability of a policy mediates the relationship between the perceived accessibility of the policy and entrepreneurs' responsive preferences.*

*2.2. Policy Perceptions and Enterprise's Response Behavior*

One of the functions of the entrepreneur is to pursue innovation in specific social practices, thereby creating a favorable environment to survive and develop in the hope of obtaining and efficiently utilizing external resources, reducing the cost of innovation, and improving innovation performance [37]. However, different policy instruments obviously have different effects on an enterprise's innovation performance due to their different positionings and objectives. The instability of innovation policy effectiveness can even inhibit the technical performance of the policy [38]. In practice, full information and historical precedents often serve as references for entrepreneurs in making decisions. However, it is not advisable for enterprises to make response decisions until after obtaining sufficient information resources in an uncertain and dynamic environment. Responsive decision making requires entrepreneurs to play a double role as both "opportunity-manager" and "risk-taker", meaning not only having a broad imagination, but also the wisdom to seize opportunities [39]. Therefore, for a policy whose incentive effect is reflected in the enterprise's innovation inputs, innovation management, and operation mechanisms, the more the entrepreneurs perceive the policy as effective, the deeper their sense of identification with it, and the more likely they are to have a positive attitude toward accepting the policy support [23,39]. Most firms choose to actively respond to innovation polices with which they are already familiar or which have proven to be practical by their partners, in order to seek new opportunities and reduce their response risk [40]. Some scholars have also found that an enterprise's favorable response to innovation policy can have a positive impact on the promotion of the enterprise's innovation performance and that different decision makers adopt different response mechanisms according to the market demands and environmental changes [41]. In addition, an Industry 4.0 policy, as a type of innovation policy, is a combination of a series of instruments, with heterogeneity of the functions, regardless of the policy target or intended diversity of entrepreneurs' responsive behaviors, which can lead to an innovation policy response process containing multiple response paths [42]. Therefore, the following hypotheses are proposed:

**Hypothesis 4 (H4).** *The perceived practicability of a policy has a positive effect on entrepreneurs' responsive preferences, meaning that the more practical the policy is, the deeper the entrepreneurs' sense of identifying with it, and the more likely they are to accept the policy guidance.*

**Hypothesis 5 (H5).** *The perceived practicability of a policy has a positive effect on an enterprise's response behavior.*

**Hypothesis 6 (H6).** *Entrepreneurs' responsive preferences mediate the relationship between the perceived practicability of a policy and an enterprise's response behavior.*

*2.3. Entrepreneurs' Responsive Preferences and Response Behavior*

Many studies show that the main driving force of innovation is to transform both innovation and entrepreneurship in significant ways [43]. An enterprise's cultural heritage of entrepreneurship, in terms of the will and courage of its decision makers, influences the enterprise's subsequent development. As the decision maker, the entrepreneur is often regarded as the key factor leading to the success or failure of the enterprise to innovate [44]. The costs involved in the initial inputs into technological innovation are large, so it is wise for enterprises to use external resources brought by their response to innovation policies to balance their innovation costs. How the enterprise responds to the innovation policy is closely related to its strategy. Under a given policy environment, an enterprise's behavior in response to an innovation policy should not be one of blindly following, but rather the result of a game of maximizing the enterprise's benefit [45]. In this kind of game, there are some key factors, such as the degree of attractiveness of the policy and the entrepreneur's sense of identifying with it, which are related to the government supply side and the enterprise demand side of technological innovation [45,46]. In addition, due to the impact of input–output mechanisms based on resources, the entrepreneurs' attitudes toward responding to policies are also related to the differences between the enterprise's innovation learning model and innovation activities [47]. Different models also correspond to strategies such as "sticky response", "relationship response", etc. [48]. The above content reflects the market competition and enterprise initiative of the innovation policy response. Therefore, we propose the following hypothesis:

**Hypothesis 7 (H7).** *Entrepreneurs' responsive preferences have a positive effect on an enterprise's response behavior, meaning that the more inclined an entrepreneur is to respond to a given innovation policy in their decision making, the more actively the enterprises' response to the innovation policy will be implemented.*

*2.4. Policy Perceptions, Entrepreneurs' Responsive Preferences and Response Behavior*

Some scholars point out that tax exemption policies lead to intense competition because of their wide range and argue that they have a threshold effect on innovation outcomes [49]. This reflects the idea that in pursuit of better ease of use and practicality, the formulation of the policy content needs to be targeted, meaning that different types of enterprises' absorptive capacity for preferential policies, as well as enterprises' actual capacity and demand, should be considered in the policy formulation process [50]. Other scholars point out that there are pressure-driven, information-driven, design-driven, and knowledge-driven antecedent factors in an enterprise's response to a policy, and that the evaluation of an enterprise's innovation policy and policy perception belongs to the category of information-driven factors [20]. In response to the innovation policy, access to policy information is asymmetric due to the differences in enterprises' abilities in terms of qualifications, industry status, intelligence analysis, empirical evaluation, forward-looking vision, and so on, which also leads to differences in terms of recourse utilization and policy practicability [51]. The process of interpreting policy information, clarifying the response threshold, evaluating the degree of supply and demand matching and its practicability, and determining whether the entrepreneurs' preferences are responsive to the policy, is often accompanied by the decision makers' perceptions and experience. In order to promote technological innovation, entrepreneurs must acquire relevant information and clarify the risks involved in responding to the innovation policy [52]. An effective policy response must be based on a full understanding the policy's content [28]. Feedback on

policy implementation effectiveness can provide inspiration for enterprises to perceive and respond to the policy [53]. We, therefore, propose the following hypotheses:

**Hypothesis 8 (H8).** *The perceived practicability of a policy mediates the relationship between the perceived accessibility of the policy and the enterprise's response behavior.*

**Hypothesis 9 (H9).** *Entrepreneurs' responsive preferences mediate the relationship between the perceived accessibility of the policy and the enterprise's response behavior.*

**Hypothesis 10 (H10).** *Both the perceived practicability of a policy and entrepreneurs' responsive preferences mediate the relationship between the perceived accessibility of a policy and the enterprise's response behavior.*

Figure 1 provides a heuristic exploration of this study.

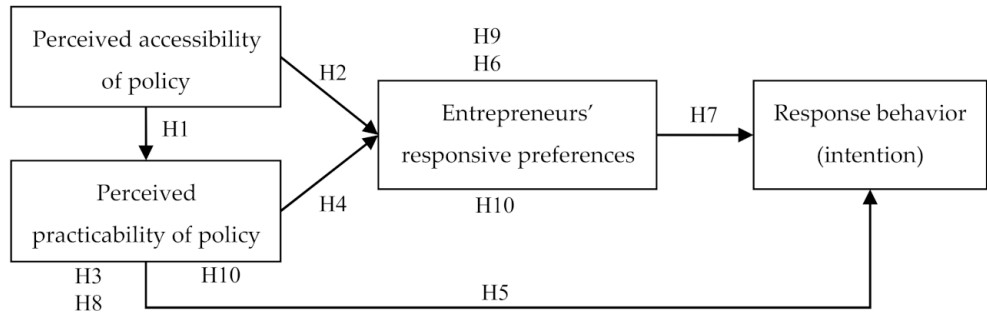

**Figure 1.** The conceptual model of this study. Source: Authors.

### 3. Research Methodology

*3.1. Questionnaire Development*

In this study, we examine enterprises' behavioral intentions to respond to Industry 4.0 innovation policies under the influence of policy perceptions and entrepreneurs' preferences. For data collection, the questionnaire was designed in two steps. First, we designed measures following a literature review and analysis and generated content validity. Following previous studies, we introduced four latent variables, namely, policy accessibility (perception), policy practicability (perception), entrepreneurs' preferences, and response behavior intention. Pierce et al. [17] suggest that the policy accessibility, in fact, means whether a policy is targeted and can easily help an enterprise or if an enterprise can correctly study and understand the policy's content. This is in line with the policy formulation goals of Hobday et al. [32] in policy content perception and interpretation. Therefore, this paper uses policy accessibility variables. Referring to the measuring methods of Pierce et al. [17], Hobday et al. [32], Havas and Weber [34], and other scholars, perceived accessibility of policy (PAP) is measured by four items, namely, policy pertinence (PAP1), content clarity (PAP2), policy response threshold (PAP3), and policy response cost (PAP4). Referring to the measuring method by Pierce et al. [17] and Jia et al. [54], perceived practicability of policy (PPP) is measured by three items, namely, the effectiveness of the enterprise's support for innovation (PPP1), policy resource allocation rationality (PPP2), and the enterprise's performance in promoting innovation (PPP3). Referring to the measuring methods proposed by Chatfield et al. [39] and Timmermans et al. [52], the entrepreneurs' responsive preferences (ERP) are measured by four items, namely, the entrepreneur's degree of sense of identity (ERP1), the enterprise's innovation and demand fit (ERP2), the enterprise's policy response experience (ERP3), and the enterprise's craving for external resources (ERP4). Referring to the studies by Pierce et al. [17], Ashford et al. [14], and other scholars, the enterprise's response behavior intention (RBI) is measured by three items, namely, policy response urgency (RBI1), policy resource demand intensity (RBI2),

and policy response profit expectation (RBI3). Therefore, the questionnaire is divided into six parts: personal information, business information, perceived accessibility of policy, perceived practicability of policy, entrepreneurs' responsive preferences and enterprise's response behavior intention, involving a total of 18 variables and 33 items. According to the suggestion by Hair et al. [55] to use at least a 5-point scale, a 7-point Likert scale (1—no agreement to 7—total agreement) is used to quantify the answers (except for personal information and enterprise information).

Second, to ensure the rationality of the questionnaire structure, the formal questionnaire was formed under an expert review, with a questionnaire pre-test and questionnaire revision. We conducted a pre-test to test the reliability and validity with the help of the Beijing Modern Manufacturing Industry Development and Research Institute and the China Scientific Association Strategy Institute, with 50 questionnaires, of which 47 were valid. High–low difference tests were conducted on all the items (the figures 27% and 73% were used as the high–low grouping critical points), and the grouping difference of all items was significant (with $p$-values of less than 0.05 and t-values greater than 1.96). Additionally, the results of factor analysis verified the overall reliability of the questionnaire, as the Cronbach's alpha of each scale (PAP, PPP, ERP, RBI) was 0.787, 0.905, 0.874, and 0.748, respectively, meaning that the scale had good reliability; at the same time, factor analysis showed that the factor load of each item was above 0.6, indicating that all latent variables were accepted.

To sum up, the questionnaire has a certain degree of reliability and validity and can be used to carry out the following investigation and research. The control items included enterprise age, enterprise size, CEO age, and CEO education level [56,57].

### 3.2. Data Collection and Sample Profile

In the environment of open innovation and Industry 4.0, technological innovation has become particularly critical to both high-tech enterprises and conventional firms' transformational enterprises [8]. The data for this study were collected via a survey of domestic Chinese high-tech enterprises located in the National Independent Innovation Demonstration Zone of China. All samples were identified by the Torch High Technology Industry Development Center (China Ministry of Science and Technology) and published on the official website http://www.innocom.gov.cn (accessed on 20 January 2021).

For randomly selected samples, two ways of distributing and collecting questionnaires were followed: one was to send questionnaires (in electronic form) to enterprise policy makers and collect them with the help of the Beijing Modern Manufacturing Industry Development and Research Institute and China Scientific Association Strategy Institute; the other was to send questionnaires (in electronic form) to trainees working on high-tech enterprise management under the support of an MBA alumni network for universities in China. Out of a total of 550 questionnaires distributed, 389 were collected and the recovery rate was about 71%. Of these, 337 questionnaires were valid, so the valid recovery rate was about 87%. In the samples, emerging strategic enterprises in the fields of software, electronic information, biotechnology, pharmaceutical manufacturing, new materials, Internet services, etc., accounted for 64% of the total, and enterprises in automobile manufacturing, machinery manufacturing, equipment manufacturing, textile, chemical, and other traditional manufacturing accounted for 36% of the total. The age of the CEOs was generally on the young side (CEOs younger than 50 years old accounted for 73%), and 61% of the CEOs reported having a postgraduate degree. The absolute value of skewness and kurtosis of each item was less than 1, which conforms to the single variable normal distribution [55]. This survey was also consistent with the suggestion of Bentler and Chou [58] that in a normal distribution, the sample number should be at least five times the parameters to be estimated, so as to be suitable for structural equation model analysis. The demographic data are presented in Table 1.

**Table 1.** Demographic information.

| Characteristics | | N | % | Characteristics | | N | % |
|---|---|---|---|---|---|---|---|
| CEO Age | <40 years | 117 | 34.7 | Level of education | Bachelor | 71 | 21.1 |
| | 41–50 years | 130 | 38.6 | | Master | 205 | 60.8 |
| | >50 years | 90 | 26.7 | | Ph.D. | 61 | 18.1 |
| Enterprise's age | <3 years | 72 | 21.4 | Enterprise's ownership | Central government | 45 | 13.4 |
| | 4–6 years | 70 | 20.8 | | State-owned | 66 | 19.6 |
| | 7–9 years | 60 | 17.8 | | Private | 166 | 49.3 |
| | >10 years | 135 | 40.1 | | Foreign joint venture | 60 | 17.8 |
| Enterprise's annual income (RMB) | <1 million | 50 | 14.8 | Gender | Female | 78 | 0.2 |
| | 100–499 million | 77 | 22.8 | | | | |
| | 500–1999 million | 97 | 28.8 | | Male | 249 | 0.7 |
| | 2000–9999 million | 59 | 17.5 | | | | |
| | >10,000 million | 54 | 16.0 | | | | |

*3.3. Data Analysis*

We used SPSS 22 for exploratory factor analysis to obtain the total scores, percentages, means, standard deviations, skewness, and kurtosis. Then, we used SEM-AMOS to analyze the data. Both the calculation model and the structural model were considered [59]. SEM-AMOS is convenient for examining cause and effect relationships between multiple independent and dependent variables, giving priority to confirming or rejecting the theories [60].

## 4. Results and Analysis

*4.1. Measurement Instruments*

First, critical ratio analysis was carried out on all the items, and the results showed that the data were suitable for factor analysis, with a KMO statistic of 0.905 and Bartlett's test of sphericity showing significance at the $p < 0.001$ level. All instruments adopted are shown in Table 2. Through factor analysis, it can be seen that all factors are significant, and the cross-factor loading is less than 0.4. Next, the correlation between each pair of items was analyzed. The results shown in Table 2 indicate that the inter-item correlation values were higher than 0.3; the values of Corrected Item–Total Correlation were higher than 0.5 and Cronbach's alpha values were higher than 0.7. Therefore, all items were identified as being suitable for factor analysis [55].

**Table 2.** Measurement model, item loadings, reliability, and validity.

| Factors | Code | Unstd. | S.E. | z-Value | p | Std. | SMC | Loading | AVE | CR | CA |
|---|---|---|---|---|---|---|---|---|---|---|---|
| Perceived accessibility of policy | PAP1 | 1.000 | | | | 0.696 | 0.484 | 0.707 | 0.504 | 0.802 | 0.801 |
| | PAP2 | 0.956 | 0.090 | 10.573 | *** | 0.707 | 0.500 | 0.813 | | | |
| | PAP3 | 1.046 | 0.095 | 11.012 | *** | 0.758 | 0.575 | 0.702 | | | |
| | PAP4 | 0.878 | 0.086 | 10.233 | *** | 0.676 | 0.457 | 0.693 | | | |
| Perceived practicability of policy | PPP1 | 1.000 | | | | 0.701 | 0.491 | 0.685 | 0.517 | 0.762 | 0.761 |
| | PPP2 | 1.162 | 0.121 | 9.615 | *** | 0.701 | 0.543 | 0.735 | | | |
| | PPP3 | 1.023 | 0.106 | 9.620 | *** | 0.737 | 0.516 | 0.778 | | | |
| Entrepreneurs' responsive preferences | ERP1 | 1.000 | | | | 0.738 | 0.545 | 0.703 | 0.509 | 0.805 | 0.805 |
| | ERP2 | 0.903 | 0.086 | 10.554 | *** | 0.666 | 0.444 | 0.713 | | | |
| | ERP3 | 1.001 | 0.091 | 10.953 | *** | 0.697 | 0.486 | 0.757 | | | |
| | ERP4 | 1.008 | 0.088 | 11.506 | *** | 0.750 | 0.563 | 0.742 | | | |
| Response behavior intention | BI1 | 1.000 | | | | 0.751 | 0.564 | 0.790 | 0.695 | 0.872 | 0.868 |
| | BI2 | 1.165 | 0.076 | 15.414 | *** | 0.912 | 0.832 | 0.822 | | | |
| | BI3 | 1.109 | 0.073 | 15.121 | *** | 0.831 | 0.691 | 0.831 | | | |

Note: *** $p < 0.001$.

We then tested the reliability and validity of the data in the survey, and the composite reliability (CR), the average variance extracted (AVE), and square multiple correlations (SMC) were used to test model convergent validity. In Table 2, the SMC values are greater than 0.5, the factor loading of each item is higher than 0.7, the CR values are higher than the cut-off value of 0.7, and the AVE values are higher than the threshold of 0.5, which proves that the convergent validity of the construct is adequate [55,61,62].

### 4.2. Reliability, Validity, and Measurement Model Interventions

SEM is usually used for confirmatory factor analysis and generating model health indices for individual health index models from measurement models to test the strength of the relationship direction [59]. The measurement elements are shown in Table 1. The results show that we have good item reliability and that the constructs have internal consistency.

The square root of AVE and the Pearson correlation tests are all examples of cross-loading. We then used cross-loading to assess discriminant validity. The comparison between the square root of AVE and Pearson correlation is shown in Table 3; the value of the diagonal and bold (square root of AVE) is greater than the value of the corresponding row and column numbers (Pearson correlations) [61]. This indicates that we have good discriminant validity. Therefore, the measurement variables are distinct from one another [63]. At the same time, correlations with other measures below 0.7 would usually be accepted as evidence of measure distinctness and thus of discriminant validity [64].

**Table 3.** Discriminant validity.

| Factors | Code | AVE | PAP | PPP | ERP | RBI |
|---|---|---|---|---|---|---|
| Perceived accessibility of policy | PAP | 0.504 | **0.710** [a] | | | |
| Perceived practicability of policy | PPP | 0.517 | 0.695 | **0.719** | | |
| Entrepreneurs' responsive preferences | ERP | 0.509 | 0.667 | 0.686 | **0.713** | |
| Response behavior intention | RBI | 0.695 | 0.543 | 0.669 | 0.621 | **0.834** |

[a] The bold numbers in the diagonal row are the square root of the AVE. Off-diagonal elements are the correlations among constructs.

### 4.3. Model Fit Evaluation

Here, we summarize the goodness-of-fit of the measurement and structural model report. As shown in Table 4, the chi-square (CMIN) and over degree of freedom (DF) (CMIN/DF) value in the measurement model was 1.585, whereas the value in the structural model was 1.566. After chi-square and DF values, the measurement model's GFI (0.954), AGFI (0.932), CFI (0.98), TLI (0.974), SRMR (0.037), RMSEA (0.042), and the structural model's GFI (0.954), AGFI (0.933), CFI (0.98), TLI (0.975), SRMR (0.038), and RMSEA (0.041) fit the criteria [63,65]. Moreover, the other most commonly reported measures of fit were less than the cut-off [66]. Therefore, both the measurement and structural model have an excellent model fit. This demonstrates that our structural model for enterprise responses to Industry 4.0 innovation policies has good adaptability.

We then employed cross-validation analysis to determine whether the model has stability and universality. Cross-validation is important not only when modifications to the original model have been undertaken following an initially poor fit, but also when the model has provided an acceptable fit in the first place [67]. The samples were randomly divided into sample groups and calibration samples, with the same sample number. As shown in Table 5, the $p$-value is greater than 0.05 and the absolute value of $\Delta$TLI is less than 0.05, indicating that there were no differences in factor loading, path coefficient, factor variance, or variable residuals. Therefore, the cross-validity assessment proves that the model accords with the congruent requirement and has stable validity.

**Table 4.** Goodness-of-fit of the measurement and structural model.

| Statistical Check | Goodness-of-Fit Criteria | Measurement Model | Structural Model | Result |
|---|---|---|---|---|
| CMIN($\chi$2) | Smaller is better | 112.503 | 112.744 | - |
| DF | Bigger is better | 71 | 72 | - |
| CMIN/DF | 1 < CMIN/DF < 5 | 1.585 | 1.566 | Good |
| GFI | >0.9 | 0.954 | 0.954 | Good |
| AGFI | >0.9 | 0.932 | 0.933 | Good |
| NFI | >0.9 | 0.947 | 0.947 | Good |
| CFI | >0.9 | 0.98 | 0.98 | Good |
| IFI | >0.9 | 0.98 | 0.98 | Good |
| RFI | >0.9 | 0.932 | 0.933 | Good |
| TLI (NNFI) | >0.9 | 0.974 | 0.975 | Good |
| PGFI | >0.5 | 0.645 | 0.654 | Good |
| PCFI | >0.5 | 0.764 | 0.775 | Good |
| PNFI | >0.5 | 0.739 | 0.749 | Good |
| SRMR | <0.08 | 0.037 | 0.038 | Good |
| RMSEA | <0.08 | 0.042 | 0.041 | Good |

**Table 5.** Cross validation.

| Model | ΔDF | ΔCMIN | *p* | ΔNFI Delta-1 | ΔIFI Delta-2 | ΔRFI rho-1 | ΔTLI rho2 |
|---|---|---|---|---|---|---|---|
| Measurement weights | 10 | 9.728 | 0.465 | 0.004 | 0.005 | −0.003 | −0.003 |
| Structural weights | 5 | 9.777 | 0.082 | 0.004 | 0.005 | 0.001 | 0.001 |
| Structural covariance | 1 | 0.005 | 0.946 | 0.000 | 0.000 | −0.001 | −0.001 |

*4.4. Structural Model and Direction Coefficient*

The effects of the independent variables by themselves on the dependent variable are reflected by the path coefficient of the structural model. The maximum likelihood method can test the complex relationship between variables that belong to different constructs, and the influence of moderating and mediating [59]. Figure 2 shows the overall results.

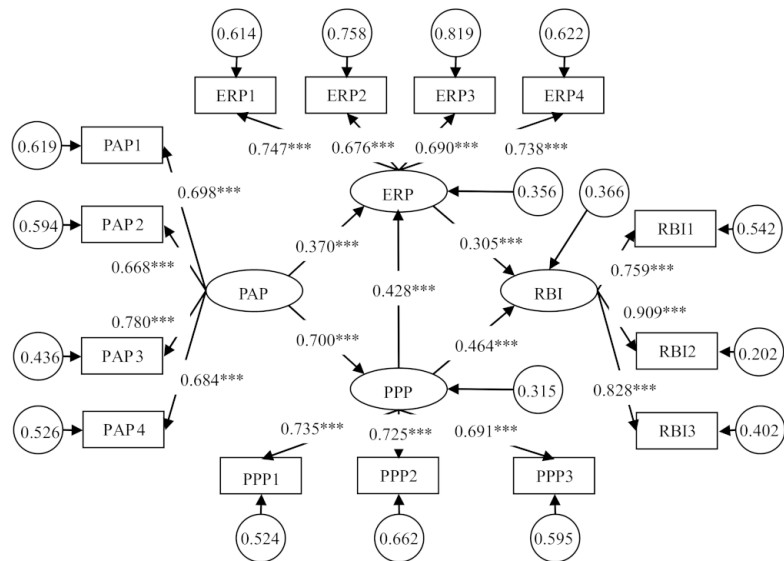

**Figure 2.** Measurement model. *** $p < 0.001$.

The path coefficients of the structural model are shown in Table 6. The regression coefficients are greater than 0.2, the R squared value is greater than 0.33, and all links were significant at the 0.001 level. This means that the path coefficients of the structural model have good explanatory power [62]. The path coefficients of perceived accessibility of policy (PAP) and perceived practicability of policy (PPP) are significant, which illustrates that PAP (= 0.700, $p = 0.001$) has a positive and important effect on PPP. This result supports Hypothesis 1. This means if the policy content is clearer to understand and easier to respond to, the perceived practicability of the policy will improve.

**Table 6.** Testing results of hypotheses.

| Hypotheses | Path | | | Unstd. | S.E. | z-Value | *p* | Std. | R $^2$ |
|---|---|---|---|---|---|---|---|---|---|
| | **Independent Variable** | | **Dependent Variable** | | | | | | |
| H1 | PAP | → | PPP | 0.808 | 0.095 | 8.510 | *** | 0.700 | 0.489 |
| H2 | PAP | → | ERP | 0.478 | 0.123 | 3.886 | *** | 0.370 | 0.540 |
| H4 | PPP | → | ERP | 0.479 | 0.109 | 4.381 | *** | 0.428 | 0.540 |
| H5 | PPP | → | RBI | 0.506 | 0.101 | 4.995 | *** | 0.464 | 0.501 |
| H7 | ERP | → | RBI | 0.297 | 0.085 | 3.472 | *** | 0.305 | 0.501 |

Note: *** $p < 0.001$.

The regression path coefficients of perceived accessibility of policy (PAP) and entrepreneurs' responsive preferences (ERP) are significant, which illustrates that PAP (= 0.370, $p = 0.001$) has a positive and important effect on ERP. This result supports Hypothesis 2. This means that CEOs are more inclined to respond to policies with clear content and low barriers to responding.

The regression path coefficients of perceived practicability of policy (PPP) and entrepreneurs' responsive preferences (ERP) are significant, which illustrates that PPP (= 0.428, $p = 0.001$) has a positive and important effect on ERP. This result supports Hypothesis 4. This means that CEOs pay more attention to whether policies have a practical effect on current enterprise innovation. CEOs will accumulate experience and increase their preference for policy responses through policy response behaviors.

The regression path coefficients of perceived practicability of policy (PPP) and response behavior intention (RBI) are significant, which illustrates that PPP (= 0.464, $p = 0.001$) has a positive and important effect on RBI. This result supports Hypothesis 5. This means that the behavioral intention of enterprises to respond to policies is determined by the practicality of the policies. If policies can bring benefits to enterprise innovation, enterprises will accept government guidance and implement responses.

The regression path coefficients of entrepreneurs' responsive preferences (ERP) and response behavior intention (RBI) are significant, which shows that ERP (= 0.305, $p = 0.001$) has a positive and important effect on RBI. This result supports Hypothesis 7. This indicates that CEOs are the corporate decision makers. They will evaluate the consequences of policy response actions based on their experience and preferences, and consequently, persuade the board of directors to respond to the policy.

As can be seen from the above results, the path coefficient of the perceived practicability of policy after standardized regression (= 0.464, $p = 0.001$) is higher than the path coefficient of entrepreneurs' responsive preferences for the impact on the enterprise's response behavior (= 0.305, $p = 0.001$). Therefore, the policy perception of practicability is more effective than entrepreneurs' responsive preferences to drive enterprises to respond to policies.

*4.5. The Mediating Effect*

We used the bootstrap method, which is being used with increasing frequency in research, to assess the mediating effects. Simulation research shows that bootstrapping is more powerful than the Sobel test and the causal steps approach to testing intervening variable effects [68,69]. The mediating effect of the Industry 4.0 innovation policy response

model is shown in Table 7. The multi-mediator model shows that, as expected, the perceived practicability of policy (PPP) plays a mediating role (= 0.387, z-value = 3.172 > 1.96, the bootstrap confidence interval does not contain 0) between perceived accessibility of policy (PAP) and entrepreneurs' responsive preferences (ERP). This result supports Hypothesis 3. Moreover, entrepreneurs' responsive preferences (ERP) play a mediating role (= 0.136, z-value = 2.061 > 1.96, the bootstrap confidence interval does not contain 0) between perceived practicability of policy (PPP) and response behavior intention (RBI). This result supports Hypothesis 6. Additionally, the perceived practicability of policy (PPP) plays a mediating role (= 0.384, z-value = 3.097 > 1.96, the bootstrap confidence interval does not contain 0) between perceived accessibility of policy (PAP) and response behavior intention (RBI). This result supports Hypothesis 8. Furthermore, entrepreneurs' responsive preferences (ERP) play a mediating role (= 0.134, z-value = 1.971 > 1.96, the bootstrap confidence interval does not contain 0) between perceived accessibility of policy (PAP) and response behavior intention (RBI). This result supports Hypothesis 9. In particular, the perceived practicability of policy (PPP) and entrepreneurs' responsive preferences (ERP) play a dual-mediating role (= 0.109, z-value = 2.019 > 1.96, the bootstrap confidence interval does not contain 0) between perceived accessibility of policy (PAP) and response behavior intention (RBI). This result supports Hypothesis 10.

**Table 7.** Test of mediation.

| Hypotheses | Path | Estimate | S.E. | z-Value | Bias-Corrected | | Percentile | |
|---|---|---|---|---|---|---|---|---|
| | | | | | Lower | Upper | Lower | Upper |
| H3 | PAP→PPP→ERP | 0.387 | 0.122 | 3.172 | 0.166 | 0.667 | 0.135 | 0.640 |
| H6 | PPP→ERP→RBI | 0.136 | 0.066 | 2.061 | 0.030 | 0.295 | 0.026 | 0.282 |
| H8 | PAP→PPP→RBI | 0.384 | 0.124 | 3.097 | 0.184 | 0.680 | 0.168 | 0.669 |
| H9 | PAP→ERP→RBI | 0.134 | 0.068 | 1.971 | 0.033 | 0.328 | 0.023 | 0.286 |
| H10 | PAP→PPP→ERP→RBI | 0.109 | 0.054 | 2.019 | 0.028 | 0.265 | 0.020 | 0.234 |
| | H8,H9,H10 Total IE | 0.627 | 0.133 | 4.714 | 0.414 | 0.933 | 0.401 | 0.921 |
| | H8,H9,H10 DE | 0.057 | 0.160 | 0.356 | −0.263 | 0.370 | −0.251 | 0.380 |
| | H8,H9,H10 Total effect | 0.684 | 0.130 | 5.262 | 0.435 | 0.958 | 0.441 | 0.960 |
| | H8 and H9 | 0.250 | 0.157 | 1.592 | −0.042 | 0.605 | −0.051 | 0.591 |
| | H8 and H10 | 0.275 | 0.143 | 1.923 | 0.004 | 0.606 | −0.005 | 0.588 |
| | H9 and H10 | 0.025 | 0.076 | 0.329 | −0.119 | 0.195 | −0.126 | 0.191 |

Note: Bootstrap 1000 times, 95% confidence intervals. DE, direct effect; IE, indirect effect.

In contrast, the direct effect (= 0.370) of perceived accessibility of policy on entrepreneurs' responsive preferences is less than the mediating effect (= 0.387) of perceived practicability of policy between them. Similarly, the direct effect (= 0.464) of perceived practicability of policy on response behavior intention is higher than the mediating effect (= 0.136) of entrepreneurs' responsive preferences between them. These analyses show that the mediating role of perceived practicability of policy (Hypothesis 3) and entrepreneurs' responsive preferences (Hypothesis 6) is partially mediated. However, the mediating effect (= 0.384) of perceived practicability of policy between perceived accessibility of policy and response behavior intention (Hypothesis 8) is completely mediated because there is no direct effect of perceived accessibility of policy on response behavior intention. Similarly, the mediating effect (= 0.134) of entrepreneurs' responsive preferences between perceived accessibility of policy and response behavior intention (Hypothesis 9) is completely mediated. Regarding Hypothesis 10, the dual-mediating indirect effect (= 0.109) of the perceived practicability of policy and entrepreneurs' responsive preferences is completely mediated, as the direct effect is not significant (in Table 7).

The above results show that the dual mediating effect is weaker than the single one and indicate that the perceived practicability of a policy has a stronger influence on an enterprise's response behavior than others.

## 5. Discussion

With this study, we aimed to achieve a better understanding of how enterprises' response behavior intentions concerning innovation policies are influenced by perceptions of policies and entrepreneurs' preferences. We considered other authors' suggestions to supplement existing theories with different variables and relationships. In particular, several issues of potential relevance to policy creators and responders have been accounted for in this study. The Policy Acceptance Model (PAM) [17,70] has been used in addition to including variables and relationships of a Dual-Agency Model of Firm CSR [71]. These variables and relationships of public agents (government policies) and private agents (corporate CEOs) have not previously been studied together in the policy response research area. Next, the theoretical and practical implications are discussed.

### 5.1. Theoretical Implications

First, the response preferences of entrepreneurs have a positive effect on the willingness of enterprises to respond to policies (=0.305). Some studies have discussed the response relationship between firms and policies [7,20,22]. However, these studies emphasize the importance of policies in improving the external environment for enterprises' innovation capabilities. Lin et al. [20] proved that "environmental-side" and "demand-side" policies are increasingly favored by policy makers. Skordoulis et al. [72] found that companies prefer policies that enhance their competitive capability. The effective implementation of Industry 4.0 innovation policies must be recognized by entrepreneurs, mainly technology companies, in order to achieve the purpose of the policy intervention and drive innovation. As a result, entrepreneurs have increased their willingness to respond to practical policies, and at the same time, have established close government–enterprise ties. The government should regard entrepreneurs' responses to policies as an important factor in policy evaluation. Industry 4.0 innovation policies can continuously improve how they are perceived (by working on issues such as clarity of policy content, policy objectives, policy targets, response thresholds, response benefits, etc.) and make them more effective.

Second, PPP is significant and positively influences enterprises' response behavior intention (= 0.464). The direct impact of PPP on entrepreneur's response preference is also significant ($\beta$ = 0.428). These analytical results reveal that policy makers can engage in "government–enterprise dialogue" to deeply understand the actual needs of entrepreneurs, thereby encouraging enterprises to respond to policies and enhancing the policies' effectiveness. Liu et al. [73] proved that companies also hope that policies can help companies reduce their costs and promote a win-win situation for supply chain partners. Consequently, improving the practicability of policies can also save public resources and reduce costs.

Third, PAP positively and significantly affects PPP (= 0.70), which exerts a considerable influence on entrepreneurs' response preferences (= 0.37). Therefore, PAP is an important indicator of Industry 4.0 innovation policies. Moreover, the test on mediating effects shows that PAP has a significant mediating impact on entrepreneurs' response preferences through PPP. Some studies [21] focus on the strictness of policy content and on policy practicality. This study reveals the empirical influence of PPP on entrepreneurs' response preferences and response behavior intentions, and, furthermore, shows that this influence is based on PAP. Aquilani et al. [74] revealed that firms seek benefits and new and/or different opportunities, which is important in shaping entrepreneurs' preferences concerning innovative decision making. Thus, this study will help technology companies seek innovation opportunities through policy guidance.

### 5.2. Practical Implications

With the continuous development of data mining technology in recent years, policy text mining and semantic analysis can help improve the perceived accessibility of policies. Policy text mining can also help entrepreneurs understand the purpose and function of policies, thus enhancing the perceived practicality of policies. This provides support

for Hypothesis 1. Therefore, improving how policies are perceived often requires the application of advanced text analysis techniques. Professional managers and decision makers help obtain external resources for enterprises by responding to government policies.

The findings show that perceptions of policies have a positive effect on the entrepreneurs' responsive preferences (Hypotheses 2 and 3), which is supported in some studies. For example, Flanagan et al. [75] proposed that policy makers often put too much faith in coordination and intelligent design of "policy mixes". This means that it is dangerous for policy makers to pay attention to the instrumentality of policies while ignoring entrepreneurs' understanding of policies. In particular, policy path dependence indirectly shows that the perceived accessibility of a policy has a stimulating effect on the preference for responding to the policy. Policy makers should fully understand the policy role of guidance, innovation-driving, resource allocation, etc., and then enhance entrepreneurs' preferences regarding the accessibility and practicability of policies through training and interpreting.

Additionally, the findings of this study reveal that for an enterprise to accept the policies formulated by the government, a positive response attitude must be encouraged and cultivated. Positive attitudes toward policy responses among entrepreneurs are influenced by their preferences. Therefore, government implementers should survey entrepreneurs' attitudes toward the changes being suggested in order to shape the co-creation of the policies. In this way, practical policies can be offered to meet the actual needs of enterprises' operations and innovation activities. This provides support for Hypothesis 4. Therefore, future work must strengthen research on policy response paths and response effects.

Furthermore, entrepreneurs' preferences positively affect their response behavior intention (Hypothesis 5), and also affect the innovation effect of the enterprise response policy [76]. A CEO's educational background and management experience are important factors that affect policy response preferences [77]. Therefore, the determination of entrepreneurs' responsive preferences is a complicated process, requiring the introduction of more variables.

Moreover, the motivation of this study is to empirically address entrepreneurs' response mechanisms during policy making and implementation. Industry 4.0 involves drastic changes in the value chain and environmental governance, involving a series of related policies in the future. The "environmental-side" dimension, which is particularly relevant to externalities, draws attention within the overall China Manufacturing 2025 program, even though not all entrepreneurs will benefit from anticipating particular policies. Accordingly, policy-based approaches are required to improve sustainable entrepreneurship. However, what is not so clear-cut is when and how the reactions of entrepreneurs will be taken.

In addition to attempting to validate the effective mediating consequences of the perceived practicality of policies and entrepreneurs' preferences, the current study suggests practical and policy-based approaches to improving sustainable entrepreneurship. Such an understanding of the resilience and sustainability of interaction between government and business may be helpful in determining the factors contributing to the robustness of the innovation system. As the environment and long-term sustainability are important prospects in terms of Industry 4.0, this study contributes to narrowing the gap between the selfishness and altruism inherent in the sustainable promotion of innovation.

## 6. Conclusions

China is the largest developing country and the second-largest economy in the world. Driven by the "Made in China 2025" plan, the Chinese government has implemented several policies to harness the potential high-tech manufacturing capacity of Industry 4.0. We find that the policy coverage of Industry 4.0 is too broad, involving 10 industry fields such as information technology, numerical control tools and robotics, aerospace equipment, ocean engineering equipment, railway equipment, new energy, power equipment, new materials, biological medicine, and agricultural machinery, etc. Admittedly, for the government's policies

to play the role of the "visible hand" in promoting innovation, this requires enterprises to respond. However, the profit-seeking nature of enterprises often leads to negative responses, especially among private enterprises [78] and SMEs [79]. Therefore, the issue of policy responses from enterprises and entrepreneurs is worthy of more in-depth study.

Our study introduces factors of policy perceptions and entrepreneurs' preferences and explores their role in policy response behavior. Through hypothesis induction and empirical analysis, the 10 hypotheses proposed in this paper have all been verified by structural equation modeling of China's enterprises' response to Industry 4.0 innovation policies. We find that the key factor determining enterprises' response to policies is the perceived practicability of policy. The cumulative effectiveness of policies in practice is an important tool for the government to drive enterprises to engage in technological innovation in the Industry 4.0 era. The perceived practicability of policies and entrepreneurs' responsive preferences are both conducive to the promotion of enterprises' response behavior. Entrepreneurs' response preferences play a partial mediating role between the perceived practicability of policies and enterprises' response behavior. Furthermore, the findings are supported by a subsequent test of mediation. First, the perceived accessibility of policy does not have a direct impact on enterprises' response behavior intention but indirectly impacts it through the dual-mediating effect of the perceived practicability of policy and entrepreneurs' responsive preferences. Second, the perceived practicability of policy plays a partial mediating role between the perceived accessibility of policy and entrepreneurs' responsive preferences. Then, the mediating effect of the perceived practicability of policy is greater than the mediating role of entrepreneurs' responsive preferences and is also greater than the dual-mediating effect of the perceived practicability of policy and entrepreneurs' responsive preferences. We also find that the response preferences of entrepreneurs will be influenced by policy perceptions. Policy response training and official policy interpretation are critical to improving entrepreneurs' response attitudes.

The results of this study also have implications for policy makers and entrepreneurs. First, improving the PPP is not only the basic goal of innovation policy formulation, but also the decisive factor for the perceptions of policies to guide enterprises to respond rationally. The PPP is closely related to the PAP. In the meantime, the targeting content of the policy formulation and the rationality of the response threshold can help enhance the PPP, which requires the policy receptor range, the combination of factors acting as incentives to innovation, the policy response channel, and the actual response cost to be fully considered in the formulation of innovation policy by the government. In addition, to avoid wasting government resources and ensure a rational response, an enterprise should have the ability to scientifically evaluate its own needs, policy supply, and policy effectiveness, helping entrepreneurs make decisions on the demand side more rationally instead of blindly following others. Second, the PAP needs to guide enterprises to respond to innovation policies through the PPP and entrepreneur's response preferences. The PAP, on the one hand, is derived from the clarity of interpretation of a policy, which requires the enterprise to focus on its ability to use policy resources in addition to the profits reaped in response to policies. On the other hand, it is also derived from an entrepreneur's experience with responding to innovation policies, meaning that using the existing efficient innovation policies can also achieve good perceptions. Furthermore, the full intermediary effect of the policy perceptions and entrepreneurs' responsive preferences factors illustrates the fact that the government does not need to consider forcing enterprises to respond during the innovation policy formulation. Instead, they need to take the policy implementation's purposes, steps, and predicted effectiveness into account. In addition, the publication of major policies should be accompanied by announcements or training sessions for entrepreneurs, so that business managers can fully recognize innovation policies and make rational decisions. Finally, effective response decisions require rational preferences and government guidance. Entrepreneurs' responsive preferences are often regarded as the key factor that determines business activities, and the entrepreneur's response behavior is also regarded as the process of the entrepreneur's cognition with his own

quality. However, the dual-mediating effect is weaker than the single one, which illustrates that entrepreneurs' responsive preference in innovation policy response sometimes does not play a reinforcing role; therefore, government guidance is required to foster rational entrepreneurs' responsive preferences, and it is particularly important to conduct an assessment of enterprises' response behavior before responding to the policy.

The contribution of this study is to apply the PAM and to add the CEO's response preference as a mediating variable to construct a comprehensive framework for analyzing the influence and effects of various factors on enterprises' response behaviors. Entrepreneurs are willing to respond to more practical policies to obtain access to the factor and capital resources critical to firm growth [9]. Therefore, entrepreneurs' response preferences may be an important factor in policy formulation and implementation. Industry 4.0 innovation policies with high PAP and PPP will have a better influence, positively affecting response attitudes and intentions to receive government guidance. This context of cause and effect can serve as a reference for the implementation of government innovation policies. Most studies focusing on Industry 4.0 innovation policies discuss the impact of the external environment on the company's intention and ability to innovate. This study adopts entrepreneurs' response preference as a mediating variable and explores the role of the human element in adopting innovative policies. Consequently, this is one of the few studies to verify the relationship between policy perception and response behavior from a stimulus–response perspective.

Although this study contributes to improving theoretical and practical knowledge about enterprises' behavior and intention to respond to Industry 4.0 innovation policy, it is not without limitations. The first limitation of our study is the selection of variables related to policy response decision making included in the proposed model, given the great diversity of variables in the literature. Second, this study enrolls CEOs of domestic Chinese high-tech enterprises in China as research subjects, which limits the study's generalizability. Further research should expand the corporate sample to verify our findings. Additionally, this work is limited to a single country. Thus, it is not conducive to a discussion of policy characteristics, entrepreneurs' preferences, and policy response behaviors across different countries or cultures.

As for future lines of study, other variables could be introduced to the model, or the research samples could be expanded to multiple countries. Moreover, future work could use interview methods to learn more about policy makers' and entrepreneurs' perspectives on responding to policies and their sustainability. Finally, future research should incorporate regional cultural aspects into this model and must consider the impact of historical experience on entrepreneurs.

**Author Contributions:** C.L., Z.Q., and T.F. were involved in the conceptualization, literature review, methodology design, investigation, data analysis, and review. Writing, C.L., Z.Q. All authors have read and agreed to the published version of the manuscript.

**Funding:** This research was funded by the Humanities and Social Sciences Fund Project of the Ministry of Education of China, grant number 20YJCZH066.

**Institutional Review Board Statement:** The study does not involve any hazards, such as the use of animal or human subjects' issue. And this work was approved by the Institutional Review Board of North China University of Technology, Beijing, China.

**Informed Consent Statement:** Informed consent was obtained from all subjects involved in the study.

**Data Availability Statement:** The data presented in this study are available on request from the corresponding author. The data are not publicly available due to restrictions of privacy.

**Conflicts of Interest:** The authors declare no conflict of interest.

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
