# Peer review of "The Role of Policy Perceptions and Entrepreneurs’ Preferences in Firms’ Response to Industry 4.0: The Case of Chinese Firms"

_sustainability, doi:10.3390/su132011352_

Round 1

Reviewer 1 Report

It is true that the variables to be studied proposed in the literature are very extensive. It is also true that in this work an important limitation is its specificity to a specific country such as China but, in both cases, I think that its indication as aspects to be improved is correct, as I would also add the question of trying to make the language with the statistical results obtained a little more fluid.

Author Response

Thank you very much for your encouraging and inspiring feedback on my work and for your constructive and helpful comments that have definitely improved the paper a great deal. I had studied all of your comments carefully and tried to incorporate all of them into the current version of this article. Please see the attachment.

Reviewer 2 Report

The present paper focuses on the relationship between policies perceptions and the enterprise positive behavior towards innovation in the era of the 4th industrial revolution (industry 4.0). The authors use a sample of Chinese high-tech firms and conclude that the policies perceptions (practicability, accessibility) have a positive effect on entrepreneur’s preferences which in turn motivate the positive behavior towards innovation.

It is an interesting paper that shows how the human element (such as a CEO) plays an important role in adopting innovative policies. I also believe that the technical analysis is robust since the variables are explained and checked for reliability.

I have some minor observations, mainly for the English language application.

1) The abstract needs some smoothing. There are some unclear and difficult to follow sentences. For example, in lines 11-13 “we found that the policy…have significant positive effect on the enterprise’s response behavior intention”. What is the meaning of response behaviour intention? Although you explain in the text, maybe you can make a brief explanation in the abstract to make the meaning clearer. Also, in line 13 I propose that you remove “And” from the beginning of the sentence and replace with something like “Moreover”. Furthermore lines 18-20 are unclear, I suggest you replace “dual” with “combined”.

2) The text needs to be more concise and be corrected for English language and some typos. For example in line 33 “thus” should be replaced with “this”. Line 39 “response preference often neglected” an “is” is missing. There are several other points where articles are missing or corrections are needed such as in lines 57, 82, 85, 171,174 etc. Therefore I suggest that a native or fluent English language user reviews and corrects the manuscript.

Author Response

(The authors gave the same response as above.)

Reviewer 3 Report

The paper entitled “What Determines the Enterprise’s Response to Industry 4.0 Innovation Policies in China: The Role of Policy Perceptions and Entrepreneurs’ Preferences” deals with a very interesting and contemporary subject which is in the spotlight of the several researchers during the last years.

The paper deals with the investigation of the factors role and enterprise’s response mechanism for entrepreneurs to intervene in Industry 4.0 innovation policies. The research is based on a sample of 337 Chinese high-tech firms.

The authors have to deal with the following amendments.

  • Paper’s title is too long. A shorter title like the following one would be clearer to understand “The Role of Policy Perceptions and Entrepreneurs’ Preferences in Firms’ Response to Industry 4.0: The Case of Chinese Firms”.
  • The authors must add a brief reference to the main conclusions or interpretations of the results in the abstract.
  • Literature review section is too short. It is proposed that the literature review is integrated into introduction section.
  • Following the above comment, introduction section must clearly contain and explain the aim of the paper. Furthermore, in this section the authors must explain why they examine the case of Chinese firms; it is obvious that this choice mainly derives from the fact that the authors live in China, however, this choice must be theoretically justified based on the relevant literature and data.
  • A major concern has to do with the sample. The authors state that “The data of this study were collected via a survey of domestic Chinese high-tech enterprises”. This reference could reveal a possible bias in the sample as the authors state that they refer to high-tech firms to examine their response to Industry 4.0. It could make sense that is possible for a high tech firm to respond better to Industry 4.0 than a more conventional firm. Thus, the authors must clearly explain why they made this choice and if they consider that there is no bias. Otherwise, they must clearly refer to this sample in the paper’s title, abstract, introduction and methodology so as to be clear.
  • The authors must further explain the choice of random sample.
  • Furthermore, is must be explain even the sample of 389 valid questionnaires is considered representative or not? How this number is calculated as a representative one? Did the authors refer to a specific calculation methodology? What about possible selection errors?
  • How do the authors evaluate the 87% response rate? Why do they think they obtained this high rate?
  • In the data analysis introduction the authors must refer to the level of significance they determined for their statistical analyses.  
  • In their discussion section the authors must compare their results with similar papers concerning Industry 4.0 and innovation so as to sharpen their own results. Such papers are the following ones:

    1. Kirner, E., Kinkel, S., & Jaeger, A. (2009). Innovation paths and the innovation performance of low-technology firms - An empirical analysis of German industry. Research Policy, 38(3), 447-458
    2. Frank, A. G., Mendes, G. H., Ayala, N. F., & Ghezzi, A. (2019). Servitization and Industry 4.0 convergence in the digital transformation of product firms: A business model innovation perspective. Technological Forecasting and Social Change, 141, 341-351.
    3. Skordoulis, M., Ntanos, S., Kyriakopoulos, G. L., Arabatzis, G., Galatsidas, S., & Chalikias, M. (2020). Environmental innovation, open innovation dynamics and competitive advantage of medium and large-sized firms. Journal of Open Innovation: Technology, Market, and Complexity, 6(4), 195.
    4. Liu, B., & De Giovanni, P. (2019). Green process innovation through Industry 4.0 technologies and supply chain coordination. Annals of Operations Research, 1-36.
    5. Aquilani, B., Piccarozzi, M., Abbate, T., & Codini, A. (2020). The role of open innovation and value co-creation in the challenging transition from industry 4.0 to society 5.0: Toward a theoretical framework. Sustainability, 12(21), 8943.

  • The authors should clearly explain their work’s contribution and novelty in conclusions section by adding at least one relevant paragraph. The same has to be done with policy implications
  • The reference to paper’s limitations is too brief. A section of possible limitations and caveats must be added.
  • Furthermore, due to the fact that “Sustainability” is a Journal oriented to the environment, it is proposed that the authors add 1-2 paragraphs in their discussion to explain how enterprise’s response to industry 4.0 innovation policies can contribute in environmental protection and sustainability.
  • Last, several language errors exist. Despite the fact that in most of the cases it is obvious what the authors intend to analyze, is referred on a wrong way. Thus, the authors should carefully proofread their manuscript to improve the use of English language. This could be done through the assistance of native speaker of English language or a professional.

Author Response

(The authors gave the same response as above.)

Round 2

Reviewer 3 Report

The authors have rigorously reviewed the paper and efficiently dealt with all the issues arised in the first round of revision. Thus the paper can be published in its present form.